# Global Challenges and Responses: Agriculture, Economic Globalization, and Environmental Sustainability in Central Asia

Altanshagai Batmunkh [1], Agus Dwi Nugroho [1,2,*], Maria Fekete-Farkas [3] and Zoltan Lakner [3]

[1] Doctoral School of Economic and Regional Sciences, Hungarian University of Agriculture and Life Sciences, 2100 Godollo, Hungary; shagai_eso@yahoo.com
[2] Faculty of Agriculture, Universitas Gadjah Mada, Yogyakarta 55281, Indonesia
[3] Institute of Agricultural and Food Economics, Hungarian University of Agriculture and Life Sciences, 2100 Godollo, Hungary; farkasne.fekete.maria@uni-mate.hu (M.F.-F.); lakner.zoltan.karoly@uni-mate.hu (Z.L.)
* Correspondence: agus.dwi.n@mail.ugm.ac.id

**Abstract:** Economic globalization (EG) accelerates very fast in Central Asia. This could cause environmental degradation, according to the environmental Kuznets curve (EKC) hypothesis. The study aims to determine how the EG of agriculture impacts environmental sustainability, and to test the EKC hypothesis on the agricultural sector in six Central Asian countries. Particularly, some main hypotheses were proposed using secondary data from Kazakhstan, Kyrgyzstan, Mongolia, Tajikistan, Turkmenistan, and Uzbekistan from 1994 to 2019. This study uses five explanatory variables: agricultural exports value (EXP), agriculture forestry and fishing value-added (AVA), the exchange rate (EXR), total natural resource rents (RENT), and external debt stocks (DEBT), while the dependent variable in this study is the $CO_2$ emissions from on-farm energy use (EMS), temperature changes (TEMP), and forest fires (FIRE). These data are analyzed using panel data regression. As a result, AVA and RENT raise EMS; EXC raises TEMP but lowers EMS; DEBT raises TEMP but can lower FIRE. Hence, we propose recommendations to improve this condition, including a clear roadmap, enhanced partnerships, and regional and international support.

**Keywords:** economic globalization; environmental sustainability; environmental Kuznets curve; $CO_2$ emissions; temperature changes; forest fire areas

## 1. Introduction

The environmental Kuznets curve (EKC) describes the relationship between economic growth and environmental degradation. According to this hypothesis, when a country's income is still low, its attention will be focused on increasing income by ignoring environmental quality problems. As a result, rising incomes will be followed by rising pollution, which will eventually fall with sustained growth. This decline was due to increased social control and government regulation [1].

At present, many countries are trying to enhance their income by participating in economic globalization (EG), particularly in industries with numerous benefits, such as agriculture [2]. EG, defined as the integration of economic activity across borders through markets, is at the heart of a comprehensive globalization phenomenon that spans economic, social, cultural, and environmental aspects [3]. Related to agriculture, EG boosts income, employment, markets, value chains, national specialties, export diversification, and modernization [4,5]. However, as revealed by the EKC hypothesis, this effort (EG) harms the environment [6]. For instance, there is an increase in emissions, temperature, deforestation, and water scarcity due to the increased demand for agricultural products [7–9]. Conservation efforts began in response to a growing awareness of environmental sustainability

among various countries and consumers [10]. EG generates the economic resources that allow for environmental stewardship and cleanup [3]. Eco-friendly agreements and food certifications have been launched to meet these goals [11].

For example, several countries promised to limit emissions and mitigate climate change in the United Nations Framework Convention on Climate Change (UNFCCC). One of the significant UNFCCC moments, binding many countries, is the Paris Climate Accord (PAC) 2015, or the 21st Conference of the Parties (COP21). The primary outcomes of COP21 are for each country to reduce greenhouse gas (GHG) emissions, encourage renewable energy production, maintain global temperatures below 2 °C, or ideally, below 1.5 °C, and donate funds to help developing countries cope with the effects of climate change [12].

There are three frameworks to achieve the COP21 outcomes: financial, technical, and capacity-building support. Developed countries must provide financial assistance to developing countries for climate change mitigation (reduce emissions) and adaptation (minimize the negative consequences of climate change). Policy instruments and their implementation must continually drive toward the development and transfer of technology. This will boost climate change resilience while also lowering GHG emissions. The last point is climate-related capacity building for developing countries, and calls on all developed countries to strengthen their support for such activities in developing countries [12].

On 1–12 November 2021, COP26 was held in Glasgow. The purpose of the conference was to assess the outcomes of COP21. The result is a strengthening of each country's commitment for achieving the COP21 outcomes. Countries reaffirm their commitment to contribute USD 100 billion in aid from developed countries to developing countries each year. Countries were also requested to phase out coal-fired power facilities and inefficient fossil fuel subsidies. Countries also finalized the PAC rulebook regarding market mechanisms, non-market measures, and the transparent reporting of climate actions, and supported supplied or received aid, including loss and damage [12].

Although the agreement's impact is quite positive, countries will require time to achieve it. Furthermore, EG has been quickly increasing for a long period, particularly in developing countries. One of the regions where EG accelerates very fast is Central Asia. According to the KOF globalization index [13], EG in this region has increased by 55% over the last four decades. Hence, many problems exist in Central Asia's ecosystem and are consistent with the EKC hypothesis. As evidence, the Central Asian region has recently been warmer and wetter than a decade ago. Thus, people in the area have begun cultivating crops (such as wheat) earlier in recent years [14].

The Central Asian region has also undergone some extreme conditions, which piqued our interest in conducting this study. First, Central Asian countries implemented reforms to transition from planned to market-oriented economies after 1990. The agricultural sector was restructured, with large-scale state-owned farms giving way to small-scale individual plots of land [15]. Hence, countries in the area are particularly eager to participate fully in EG. Second, Central Asia faces a major environmental issue regarding industrial and mining activity, including radioactive soil and water contamination from uranium tailings, and chemical pollution from big industrial units. On the other hand, Central Asian countries have big ambitions to achieve a livable and sustainable future by 2050. They have started several initiatives, including achieving zero net national carbon and other greenhouse gas emissions, increasing renewable energy sources to 50% or more, and lowering city carbon footprints [16].

Third, the Belt and Road Initiative (BRI) was built in Central Asia to increase economic development. It is expected to contribute to the economic, social, and environmental components of their sustainable development goals (SDGs), both directly and indirectly [17,18]. Central Asia has a vital role in the BRI's development since it connects China with Europe, Africa, East Asia, Russia, and the Middle East [19]. However, this may influence the region's escalating environmental harm. Thus, a better understanding of environmental sustainability in Central Asia is crucial for the region's long-term economic development and environmental conservation [20]. Fourth, natural resource interdependence exists

among Central Asian countries. If one country's natural resources are harmed, it will cause problems in other countries. For example, water resources are increasingly under stress in Central Asia, as downstream countries become increasingly reliant on upstream countries [21]. Thus, irrigation is becoming increasingly limited due to climate change and human activity. This will reduce agricultural production and disrupt food security.

Agriculture was chosen as the subject of this study because it is highly vulnerable to climate change and water scarcity [22]. The most important crops in Central Asia are wheat, barley, maize, potatoes, oilseeds, and a range of vegetable crops. Meanwhile, animal husbandry is now dominated by sheep, goats, cows, and horses. Agriculture is still the most important source of employment in Central Asia and accounts for roughly a third of its gross domestic product [23,24]. People who live in rural areas of Central Asia and work in other sectors frequently do subsistence farming in addition to their off-farm employment [25]. Agriculture is also important because it provides (1) food, which may influence the country's economic growth, (2) income and foreign exchange profits, (3) overhead investment and secondary industrial expansion, and (4) rural net cash income as a stimulus to industry [26].

However, agriculture in Central Asia still faces many issues. The most major fundamental issues when adopting new technologies and mechanization into the sector are workers' low incomes, a lack of capital, the small-scale character of businesses, the farmers' low level of education and training, and poorly developed infrastructure. Central Asian countries rely on imports of production materials, such as artificial fertilizer, seeds, and fuel (oil) [27].

Central Asian agriculture uses roughly 60% of the region's water. As a result, agriculture will be hampered by environmental harm, particularly water scarcity or pollution [28]. This is exacerbated by inefficient water use owing to the deterioration of irrigation canals, excessive evaporation, and drought, while flooded arable land is frequently afflicted by salinization. Central Asian agricultural sectors also suffer from erosion and soil degradation caused by natural disasters such as droughts and floods and excessive chemical inputs [27]. According to the World Bank [29], temperatures in the region will be 1.9 °C to 2.4 °C higher in 2050. Diseases and plagues will be more likely in both agriculture and animal husbandry. This will possibly result in a significant reduction in agricultural yield per hectare.

This is certainly interesting because EG can stimulate economic growth. On the other hand, EG has the potential to harm the environment. While the Central Asian countries have agreed to abide by the UNFCCC agreement, based on the various conditions that we have previously disclosed, our study question is as follows:

RQ1: What is the impact of EG in agriculture on environmental sustainability in Central Asia?

RQ2: Does EKC apply to agriculture development in Central Asia?

Finally, this study aims to determine the impact of EG in agriculture on environmental sustainability and test the EKC on the agricultural sector in six Central Asian countries. The novelty of our study is using total natural resource rents as one of the determinant factors of environmental sustainability. This study is critical for realizing several SDG agendas: grow affordable and clean energy, organize climate action, develop life below water, and advance life on land [30]. Furthermore, according to the third national communication to the UNFCCC, climate change resilience is primarily addressed by focusing on economic growth, population welfare, poverty reduction, economic diversification, communication infrastructure, and political stability [16].

The remainder of this study is organized as follows. Section 2 introduces the previous literature on the progress of EG and environmental sustainability in Central Asia, as well as the framework of this study. Section 3 describes the data and specifies the empirical model. Sections 4 and 5 present the results and discussion of this study. Section 6 discloses the conclusions, including policy implications.

## 2. Literature Review

Following the collapse of the Soviet Union in 1990, the Mongolian economy and the economies of all Central Asian countries have faced difficult times due to economic reforms. They all had different strategies to save the economy throughout the transition from a planned to a market economy. Privatization was very restricted in Turkmenistan and Uzbekistan. Due to its favorable external environment, Uzbekistan was the only country in the post-Soviet regime to raise their GDP [31]. Although privatization was Mongolia's, Tajikistan's, and Kyrgyzstan's key economic policy, the agricultural sector remained the economy's backbone [32]. Turkmenistan and Kyrgyzstan made up the Russian Federation's "agricultural basket" during the Soviet era, and this remains the case today, with the agricultural sector dominating the Kyrgyz economy [33]. However, deforestation, overfishing, global warming, air pollution, and water supply are all evident environmental challenges in agriculture-dominated countries.

### 2.1. Economic Globalization in Central Asia

There are several different forms of EG employed today, including export and import, economic reform policy, joining the trade agreement or region, the exchange rate, foreign direct investment, debt, and other activities or policies [11,34–37]. These various activities have increased FDI, trade volumes, technology, foreign tourists, international events, and reduced poverty, income inequality, hunger, inflation, and illegal economic activity in developing countries [36,38–40]. However, this is completely debatable because EG may also lead to adverse effects, such as broader income disparities, employment instability, and economic vulnerability [41]. Various protection policies are also developing in almost all countries.

Related to EG in Central Asia, Turkmenistan, Kyrgyzstan, Kazakhstan, Tajikistan, Uzbekistan, and Mongolia experienced their lowest economic levels after the Soviet Union collapsed. In addition, Mongolia was controlled by and wholly dependent on the Soviet Union. Even though their economy was improving steadily, more than 60% of the population was still living in poverty [42]. Then, Central Asian countries decided to liberalize their economies, or EG, to speed up the development of their economic conditions. The impact of globalization on the world economy has positive effects such as reducing financial costs and transactions, creating a new competitive advantage, and increasing human capital [43].

Since 1993, the leaders of the "five stans" have gathered and established "Five Central Asian States on a Common Market" to trade with one another instead of exporting to other countries. As a result, Kazakhstan and Uzbekistan's economies grew 2.5-times faster [44]. Post-Soviet nations, including Mongolia, rely on foreign aid, foreign direct investment (FDI), commercial loans, and portfolio investments to control their political opponents and maintain authoritarian regimes.

Central Asian countries have abundant natural resources. Mongolia, rich in gold and copper, had a 20-fold rise in FDI from 1997 to 2016 [43]. However, the abundance is not necessarily associated with the amount of FDI received by each country. Despite having vast natural resources such as gold, gas, and uranium, Uzbekistan and Turkmenistan have the lowest levels of FDI in the region. In contrast, Kazakhstan, having less natural resources than the other two countries, receives a massive amount of FDI to boost its economy and legitimate its authoritarian government [45]. Kyrgyzstan has received less FDI among Central Asian countries despite having abundant natural resources. All FDI is, likewise, debatable as to whether it helps increase GDP or favors foreign capital domination [46].

Furthermore, several forms of trade collaboration, notably, with China, imply EG in Central Asia. The creation and expansion of the BRI has increased Central Asian countries' commercial, transportation, and communication relations with China. People from Kazakhstan, Kyrgyzstan, Tajikistan, Turkmenistan, and Uzbekistan have emigrated to China, Russia, and Iran due to this infrastructure development [47]. Collaboration between Central Asian countries and China is also becoming wider, including military alliances, oil

trade projects with Kazakhstan and Turkmenistan, hydroelectric power plants in Tajikistan, and Central Asian countries with primary access to the Chinese market [48].

The agricultural sector in Central Asia also experienced EG and started to transform in the early 1990s and is still changing now. They included: (1) price liberalization and the removal of direct government participation in agricultural decision making; (2) land reform and farm reorganization; (3) the establishment of market and collective-action institutions [49]. These agricultural reforms resulted in more varied production systems, considerable changes in cropping patterns, and a rise in the amount of land allocated to food crops. This helps to improve food security by increasing the amount of food availability per capita [24]. Agriculture's contribution to GDP rose as well, with Turkmenistan accounting for 12.7%, Uzbekistan—17.9%, Kazakhstan—5.32%, Kyrgyzstan—14.6%, Tajikistan—23.3%, and Mongolia—12.06%, respectively, in 2020.

### 2.2. Environmental Sustainability in Central Asia

There are various indicators of environmental sustainability, including $CO_2$ emissions, heat waves, climate change or temperature, natural disasters, and forest fires [16]. $CO_2$ emission is a vital indicator of environmental sustainability. $CO_2$ emissions have increased in Central Asia due to increased population and economic output [50]. For example, the lack of central heating systems in homes and modern energy infrastructures leads to excessive use of raw coal and less than 1% of renewable energy, resulting in air pollution [51,52]. Meanwhile, Central Asia's most prominent climate changes were marked by decreasing temperature, growing glaciers, increasing precipitation, and increasing humidity during transitions from the Sub-Boreal to Sub-Atlantic period and from the Medieval Warm period to the Little Ice Age [53]. Since the twenty-first century, climate change in Central Asian countries has harmed net ecosystem productivity [20]. However, climate changes still have a positive impact, such as warmer winters and increasing precipitation in Buqtyrma River Basin areas in Kazakhstan [54].

Rainfall activity is another part of climate change's harmful influence. Rainfall is projected to drop each year. Not only rainfall, but also clean water might be in short supply or scarce due to deforestation, improper use of water resources, and urbanization [55]. Central Asia has experienced a dramatic drop in the groundwater level and increased chemical pollution of surface water and soil salinization [56]. Agriculture, industrial and residential wastewater, and solid wastes were not adequately treated, posing an environmental threat [16]. Surface water and groundwater pollution harm human health and worsen the ecological environment [56]. Meanwhile, fire occurrence frequency is relatively high in Central Asia. Fortunately, in the last five years, the fires in grassland areas have declined [57].

There are several causes for these problems, including (1) crises in economic policy that do not account for the cost of services to the environment, (2) a lack of community and public sustainability education, (3) an inability to adapt to new challenges, and (4) technological constraints [58]. Hence, regional and national sustainable development policy and a better understanding of climate change is required in Central Asia [20,59]. There are various factors to keep the environment sustainable: (i) reduce environmental pollution and enable model energy; (ii) expand the use of recycling facilities and renewable energy sources; (iii) give financial support to boost renewable technologies; (iv) optimize the global trading environment with mutual compromise and participation; (v) improve the ecological laws and public education [59,60].

### 2.3. Theoretical Framework and Hypothesis

The EKC hypothesis was proposed by Grossman and Krueger. The EKC hypothesis shows a higher economic and income growth contribution to environmental degradation [61]. As a result of the process, environmental degradation will occur via land, water, and air pollution. This is because the state will prioritize expanding production over environmental concerns. However, at a certain point, economic and income growth will

minimize environmental degradation (turning point). This results from increased environmental public awareness, technological advances, and the shift to a service-based economy [62].

The EKC hypothesis uses an inverted U-curve model to explain the relationship between economic growth and environmental degradation. There are three stages of this, namely, the pre-industrial economy (agriculture), the industrial economy (the transition from agriculture to industry), and the post-industrial economy (a service-based economic system). Environmental damage tends to increase due to changes in the economic structure of urbanization and the transition from agriculture to industry. This activity is for mass production and for meeting consumption growth. This then declines with the change in the economic structure from energy-based industries to technology-based industries and services [61,63].

The EKC hypothesis becomes a fascinating study topic, along with the strengthening of global environmental degradation. According to Camci-Cetin et al. [64], the EKC hypothesis applies in high-income countries. This study supports the hypothesis that the EKC in the area has an inverted U-curve in the long run. This may also happen in developing countries in the future, including the Central Asia region. On the one hand, agriculture is quite important in this area (first stage of the EKC hypothesis). On the other hand, when EG is implemented, the region is transforming from agriculture to industry (second stage of the EKC hypothesis). Several existing works on the relationship between economic growth and environmental degradation also consider the effect of sectoral value-added structural changes, and structural changes that occurred by international trade and globalization effects on economic growth–environment relations [65].

EG is classified into two categories in this study: trade globalization (TG) and financial globalization (FG). Agricultural exports and agricultural value-added are the two crucial TG variables used in this study. Exchange rates, total natural resource rents, and external debt stocks are all included in FG. These various variables will boost the economic growth of Central Asian countries.

Developing countries, such as Central Asian countries, mainly depend on external debt due to a lack of savings, low incomes, and weak institutional systems. Therefore, developing countries consider external debts as a significant source of economic growth [66]. However, the EKC hypothesis states that economic growth will increase environmental degradation. For example, external debt has some positive effects on short-term economic growth, but it has a detrimental impact on the environment since it leads to $CO_2$ emissions and deforestation, as the EKC hypothesis predicts [67]. According to economic theories, the effect of external debt on the economy is debated. While by classical economics, the external debt effect has a positive impact on economic growth only in the short run, the Keynesians considered the positive effects on short-run and long-run economic growth due to investments created from the external debt. Some researchers pointed out the question of efficiency of investments supported by foreign debt and its structural effects [68]. Furthermore, another study confirmed the validity of the EKC models by analyzing the correlation between economic development and environmental quality in the Czech Republic during the transition period and afterward [68,69].

We developed a theoretical framework for this study based on the EKC hypothesis, as shown in Figure 1.

Presently, almost every country globally is heavily reliant on agricultural exports. Agriculture, horticulture, and forestry exports are critical sectors in New Zealand, accounting for 49% of its GHG emissions [70]. In the same case, agriculture trade patterns between the US and other countries result in a global reallocation of land use and an increase in global GHG emissions [71]. Lee and Zhang [72] also revealed that agricultural trade liberalization boosts carbon emissions. Aside from increased emissions, agricultural exports in Mongolia also contribute to climate change and rising temperatures [73]. This situation may worsen due to low institutions and adaptation [74]. Finally, an increase in agricultural exports and infrastructure provisions causes an expansion in forest fires and deforestation [75]. On the

other hand, climate change has the opposite impact, producing a drop in exports in many nations [76].

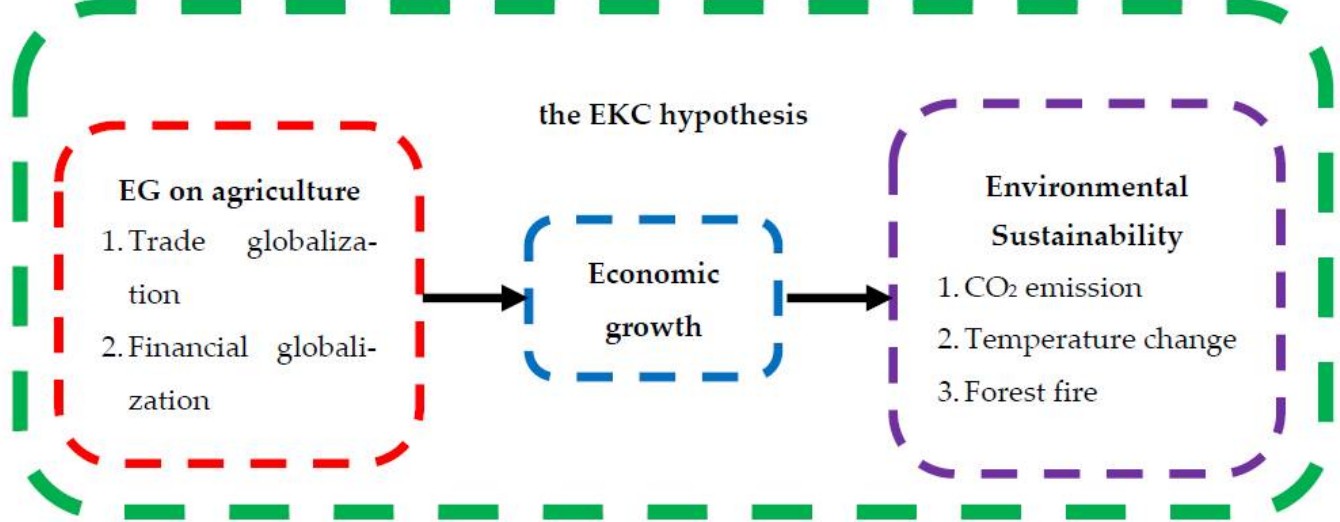

**Figure 1.** Theoretical framework of this study.

**Hypotheses 1a (H1a).** *Agricultural exports' value will increase $CO_2$ emissions from on-farm energy use.*

**Hypotheses 1b (H1b).** *Agricultural exports' value will increase temperature changes.*

**Hypotheses 1c (H1c).** *Agricultural exports' value will increase forest fires.*

There is a significant relationship in the short and long term between the increase in carbon emissions with the agricultural value-added in Turkey [77] and China [78]. A similar situation happened in Malaysia, where increased economic development (gross domestic product, financial development, industrial value-added, agricultural value-added, and manufacturing value-added) resulted in a rise in $CO_2$ emissions and the country's average temperature [79]. At the same time, Abbasi et al. [80] demonstrate that AVA has severe environmental consequences, including higher $CO_2$ levels and deforestation due to forest fires.

**Hypotheses 2a (H2a).** *Agriculture, forestry, and fishing value-added will increase $CO_2$ emissions from on-farm energy use.*

**Hypotheses 2b (H2b).** *Agriculture, forestry, and fishing value-added will increase temperature changes.*

**Hypotheses 2c (H2c).** *Agriculture, forestry, and fishing value-added will increase forest fires.*

The exchange rate harms the domestic carbon trading market when the currency value is higher and affects it positively when the currency value is lower [81]. According to Tol [82], exchange rates have a harmful influence on the environment. When a country's currency depreciates, the likelihood of climate change increases, particularly in terms of $CO_2$ and temperature. This occurs due to increased export and domestic economic activities, resulting in greater emissions and pollution. The country also becomes warmer and more reliant on agriculture as the real exchange rate depreciates [83], as does the pace of decreases in forest cover due to fires [84].

**Hypotheses 3a (H3a).** *Exchange rate will increase $CO_2$ emissions from on-farm energy use.*

**Hypotheses 3b (H3b).** *Exchange rate will increase temperature changes.*

**Hypotheses 3c (H3c).** *Exchange rate will increase forest fires.*

The natural resource rent has an impact on environmental sustainability. In the short and long run, there is a considerable effect, where a country's total natural resource rent increases $CO_2$ emissions in Saudi Arabia [85]. In Sub-Saharan African countries, the natural resource rent raises $CO_2$ emissions [86] and other pollutant emissions [87] over time. The same thing happened in the USA; total natural resource rents will put more pressure on the environment, such as climate changes and temperature increases [88]. Total natural resource rents also have negative impacts, leading to the rapid decrease of forest cover (forest fires) [89].

**Hypotheses 4a (H4a).** *Total natural resource rents' value will increase $CO_2$ emissions from on-farm energy use.*

**Hypotheses 4b (H4b).** *Total natural resource rents' value will increase temperature changes.*

**Hypotheses 4c (H4c).** *Total natural resource rents' value will increase forest fires.*

$CO_2$ emissions are influenced by foreign debt stocks. In Turkey, for example, a growth in foreign debt stocks creates a short-term increase in $CO_2$, but not in the long run [67]. External debt stocks also make countries more vulnerable to climate change, particularly temperature changes [90]. Finally, external debt stocks have a two-fold environmental impact. On the one hand, there is an increase in environmental pressure. As a result of the fire, the forest area shrinks. On the other hand, this money is employed to enhance the size of the forest by conserving it [91].

**Hypotheses 5a (H5a).** *External debt stocks' value will increase $CO_2$ emissions from on-farm energy use.*

**Hypotheses 5b (H5b).** *External debt stocks' value will increase temperature changes.*

**Hypotheses 5c (H5c).** *External debt stocks' value will increase forest fires.*

## 3. Materials and Methods

### 3.1. Data Source

The secondary data for this study were collected from six countries from 1994–2019. These countries are located in Central Asia, namely, Kazakhstan, Kyrgyzstan, Mongolia, Tajikistan, Turkmenistan, and Uzbekistan. The countries were chosen based on their Central Asian geographical location and economic situation. Other reasons were explored in Section 1.

This study uses five explanatory variables: agricultural exports value, agriculture forestry and fishing value-added, exchange rate, total natural resource rents, and external debt stocks, while the dependent variable in this study is $CO_2$ emissions from on-farm energy use, temperature changes, and forest fires (Table 1). The argument for including this variable is that agriculture contributes to several environmental issues that impact environmental degradation, such as climate change, greenhouse gas emissions, dead zones, and more.

**Table 1.** Variables and data sources of the study.

| Variable | Symbol | Source |
|---|---|---|
| $CO_2$ emissions from on-farm energy use (kilotons) | EMS | FAO |
| Temperature changes (°C) | TEMP | FAO |
| Forest fires (ha) | FIRE | FAO |
| Agricultural exports value (000 US$) | EXP | FAO |
| Agriculture, forestry, and fishing, value-added (current USD) | AVA | World Bank |
| Exchange rates | EXC | FAO |
| Total natural resource rents (% of GDP) | RENT | World Bank |
| External debt stocks (current US$) | DEBT | World Bank |

*3.2. Data Analysis*

Static panel data regression analysis was employed in this study. We chose this method because we utilized the combination of time-series and cross-sectional data. There are three static panel methods: pooled effect model (PEM), fixed effect model (FEM), and random effect model (REM) [92]. The PEM looks at how the dependent variable and some explanatory variables remain constant over time. Individual data are pooled without consideration for individual variations, resulting in a model with varying coefficients. The FEM allows for different intercepts for each cross-sectional unit but assumes that the slope coefficient is constant throughout them. Meanwhile, the lack of FEM to incorporate relevant explanatory variables that do not change over time (and possibly others that do change over time but have the same values for all cross-sectional units) results in the REM [93].

We have three dependent variables in our study, each analyzed using static panel regression. We use the log form to produce the best estimation results [92]. First, the factors that affect $CO_2$ emissions in Central Asia were estimated with the following function:

$$EMS = f(EXP,\ AVA,\ EXC,\ RENT,\ DEBT). \tag{1}$$

Based on Function (1), we formulated the static panel model:

$$log(EMS) = \beta_0 + \beta_1 log(EXP) + \beta_2 log(AVA) + \beta_3 log(EXC) + \beta_4 RENT + \beta_5 log(DEBT). \tag{2}$$

Second, the following function estimates the statistical relationship between economic globalization and temperature changes in Central Asia:

$$TEMP = f(EXP,\ AVA,\ EXC,\ RENT,\ DEBT). \tag{3}$$

The static panel model for Function (3) is:

$$TEMP = \beta_0 + \beta_1 log(EXP) + \beta_2 log(AVA) + \beta_3 log(EXC) + \beta_4 RENT + \beta_5 log(DEBT) \tag{4}$$

Third, the relationship between economic globalization and forest fires in Central Asia is depicted by the following function:

$$FIRE = f(EXP,\ AVA,\ EXC,\ RENT,\ DEBT). \tag{5}$$

Function (5) is transformed into a static panel model as follows:

$$FIRE = \beta_0 + \beta_1 log(EXP) + \beta_2 log(AVA) + \beta_3 log(EXC) + \beta_4 RENT + \beta_5 log(DEBT). \tag{6}$$

Several steps need to be taken before applying the analysis. For the first step, it is necessary to apply the unit root tests. The Levin–Lin–Chu (LLC) [94], Im, Pesaran and Shin (IPS) [95], Augmented Dickey–Fuller (ADF) Fisher Chi-square, and Phillips–Perron (PP) [96] methods are used to evaluate the stationarity of the variables. This step is required because time series data is particularly prone to spurious regression caused by non-stationary data. According to Liker et al. [97], non-stationary data can exist in the regression model, necessitating the use of a unit root test to solve the problem.

In the second step, three tests are used to evaluate the model in panel data analysis, namely, the Chow, Hausman, and Lagrange multiplier (LM) tests [98]. The Chow test may be used to see if two groups have different multiple regression functions [99]. Gregory Chow introduced this test, which is the F-test for the equivalence of two regressions. The Chow test is used to see a difference in each variable's intercept indicator ($\theta$) and interaction. If no differences exist, the data can be pooled into a single sample without taking, for differing accounts, slopes or intercepts.

The hypothesis of the Chow test is as follows:

$H_0$: $\theta_1 = \ldots = \theta_n = 0$, pooled effect model,
$H_1$: $\theta_1 \neq \ldots = \theta_n \neq 0$, fixed effect model.

The test statistic for the hypotheses is:

$$F = \frac{(SSE_R - SSE_U)/J}{SSE_U/(N-K)},$$ (7)

where $SSE_R$ is the sum of squares residuals of the restricted model, $SSE_U$ is the sum of squares residuals of the unrestricted model, $J$ is the number of restrictions, $N$ is the number of observations, and $K$ is the number of coefficients in the unrestricted model.

Hausman tests function to check for a correlation between the explanatory variable and the error term ($\rho$). The hypothesis of this test is as follows:

$H_0$: $\rho = 0$, random effect model,
$H_1$: $\rho \neq 0$, fixed effect model.

The Hausman test may be conducted with specific coefficients, using a *t*-test, or jointly, using an F-test or a Chi-square test. The test statistic for the hypotheses is:

$$t = \frac{b_{FE,\,k} - b_{RE,k}}{[var\,(b_{FE,\,k}) - var\,(b_{RE,k})]^{1/2}} = \frac{b_{FE,\,k} - b_{RE,k}}{\left[se\,(b_{FE,\,k})^2 - se(b_{RE,k})^2\right]^{1/2}},$$ (8)

where $\beta_k$ is the parameter of interest, $b_{FE,\,k}$ is the fixed effects estimate, and $b_{RE,k}$ is the random effects estimate.

The LM test, or Breusch–Pagan test for heteroskedasticity is based on a variance function ($\beta$). The general form for this function is:

$$var(y_{it}) = \sigma_\mu^2 = E\left(\mu_{it}^2\right) = h(\beta_0 + \beta_1 X_{1it} + \ldots + \beta_5 X_{5it}).$$ (9)

The null and alternative hypotheses for the heteroskedasticity test based on the variance function are:

$H_0$: $\beta_1 = \beta_n = 0$, pooled effect model,
$H_1$: $\beta_1 \neq \beta_n \neq 0$, random effect model.

The test statistic for the hypotheses is the sample size multiplied by $R^2$, and has a Chi-square ($X^2$) distribution with $S - 1$ degree of freedom [100].

$$X^2 = N \cdot R^2 \sim X^2_{(S-1)}$$ (10)

As a result of the three tests, the type of static data panel employed in this study can be decided on.

## 4. Results

Each variable in this study has a different mean and standard deviation (Table 2). This is because the variables used in this study are either in log form or in their original form.

**Table 2.** Descriptive statistics of the variables in this study.

| Variable | Mean | Std. Dev. |
|---|---|---|
| EMS, log | 2.76 | 0.60 |
| TEMP | 1.21 | 0.63 |
| FIRE | 4381.93 | 17,591.18 |
| EXP, log | 5.51 | 0.47 |
| AVA, log | 3.21 | 0.50 |
| EXC, log | 1.68 | 1.25 |
| RENT | 15.98 | 16.17 |
| DEBT, log | 3.67 | 0.64 |

According to the correlation analysis, there were no cases of multicollinearity in the independent variables (Table 3). This can be seen from the smaller correlation coefficient value of 0.8. Meanwhile, we did not analyze the correlation between the dependent variables because each of these variables was not analyzed in the same model.

**Table 3.** Correlation analysis of the variables in this study.

| Variable | EMS, log | TEMP | FIRE | EXP, log | AVA, log | EXC, log | RENT | DEBT, log |
|---|---|---|---|---|---|---|---|---|
| EMS, log | 1 | *) | *) | 0.602 | 0.759 | 0.026 | 0.005 | 0.444 |
| TEMP | *) | 1 | *) | 0.142 | 0.046 | 0.346 | 0.243 | 0.273 |
| FIRE | *) | *) | 1 | −0.472 | −0.558 | −0.439 | −0.508 | −0.692 |
| EXP, log | 0.602 | 0.142 | −0.472 | 1 | 0.628 | 0.644 | 0.342 | 0.483 |
| AVA, log | 0.759 | 0.046 | −0.558 | 0.628 | 1 | 0.488 | 0.260 | 0.762 |
| EXC, log | 0.026 | 0.346 | −0.439 | 0.644 | 0.488 | 1 | 0.347 | 0.782 |
| RENT | 0.005 | 0.243 | −0.508 | 0.342 | 0.260 | 0.347 | 1 | 0.529 |
| DEBT, log | 0.444 | 0.273 | −0.692 | 0.483 | 0.762 | 0.782 | 0.529 | 1 |

*) the relationship between the dependent variable is not analyzed.

First of all, we performed the LLC, IPS, ADF, and PP unit root tests to determine the stationarity of the data. This test is critical since the data in this study has both cross-sectional and time-series characteristics. The unit root test results in Table 4 show that not all of the dependent and explanatory variables are stationary at the level. Based on the panel data unit root tests method, TEMP, FIRE, and EXC are integrated for order zero. However, the remaining five variables are integrated for order one.

**Table 4.** Unit root test results for all variables in the model.

| Variable | LLC | | IPS | | ADF | | PP | |
|---|---|---|---|---|---|---|---|---|
| | At Level | 1st Difference | At Level | 1st Difference | At Level | 1st Difference | At Level | 1st Difference |
| EMS, log | 2.213 | −3.271 *** | 1.390 | −3.946 *** | 10.536 | 40.424 *** | 10.180 | 81.924 *** |
| TEMP | −3.020 *** | | −3.374 *** | | 31.651 *** | | 100.958 *** | |
| FIRE | −3.234 *** | | −1.812 * | | 19.814 v | | 28.600 ** | |
| EXP, log | −0.549 | −4.845 *** | −1.095 | −6.569 *** | 17.272 | 63.048 *** | 15.702 | 102.456 *** |
| AVA, log | −0.166 | −6.689 *** | 1.513 | −7.509 *** | 4.106 | 72.810 *** | 3.181 | 84.264 *** |
| EXC, log | −1.295 v | − | −1.959 * | − | 25.316 * | − | 92.769 *** | − |
| RENT | −1.836 * | −8.179 *** | −1.495 v | −7.589 *** | 18.819 v | 74.146 *** | 13.787 | 100.298 *** |
| DEBT, log | −2.032 ** | −3.870 *** | −0.051 | −3.952 *** | 13.001 | 38.191 *** | 13.262 | 45.692 *** |

Significant codes: 0 '***' 0.001 '**' 0.01 '*' 0.05 'v' 0.1, Source: author's computation (2021).

Since non-stationarity may lead to spurious regression, non-stationary variables were transformed into a stationary series before being used in panel regression analysis. As a result, five non-stationary variables were first differenced into stationary variables.

Afterward, we must choose the best model for Equations (2), (4), and (6). The Chow, Hausman, and LM tests are the answer. Table 5 summarizes the findings of the three tests. The Chow test result for Equation (2) gives a *p*-value <0.001, indicating that the rejection of $H_0$ or FEM is the preferred model. However, the Hausman test result must be conducted to determine the best model for Equation (2). The Hausman test gives a probability *p*-value <0.001 or a rejection of $H_0$; thus, we double-checked that FEM was the proper model for Equation (2).

**Table 5.** The Chow, Hausman, and LM test results for all variables in this study.

| Dependent Variable | Chow | Hausman | LM |
|---|---|---|---|
| Equation (2): CO$_2$ emissions from on-farm energy use | 106.257 *** | 143.267 *** | - |
| Equation (4): Temperature change | 8.826 | 8.424 | 130.127 *** |
| Equation (6): Forest fire | 25.354 *** | 25.597 *** | - |

Significant codes: 0 '***' 0.001, Source: author's computation (2021).

The same findings may be seen in Equation (6), where the Chow and Hausman tests provide a *p*-value < 0.001 or a rejection of Ho. Hence, the best model for Equation (6) is FEM. On the other hand, Equation (4) shows the opposite result. The Chow and Hausman tests had a *p*-value > 0.1 or failed to reject $H_0$. For that, we need to do an LM test. The result is *p*-value < 0.001, so the best model is REM.

The result in Table 6 revealed that all explanatory variables simultaneously affect the dependent variable in Equations (2), (4), and (6). This can be seen from the significant F-statistic value in each analysis. The F-test value was statistically significant at a 1% level of significance.

**Table 6.** Panel data regression in this study.

| Variable | EMS, log (FEM) | | TEMP (REM) | | FIRE (FEM) | |
|---|---|---|---|---|---|---|
| | Coef. | Std Error | Coef. | Std Error | Coef. | Std Error |
| EXP, log | −0.093 (−0.910) | 0.102 | −0.263 (−1.453) | 0.181 | 3548.170 (0.431) | 8227.749 |
| AVA, log | 0.235 ** (2.700) | 0.087 | −0.063 (−0.350) | 0.180 | 9921.253 (1.413) | 7020.266 |
| EXC, log | −0.282 *** (−4.891) | 0.0567 | 0.091 * (2.051) | 0.045 | 4867.206 (1.064) | 4574.123 |
| RENT | 0.132 * (2.139) | 0.062 | 0.150 (1.618) | 0.092 | −5504.906 (−1.102) | 4997.211 |
| DEBT, log | 0.047 (0.706) | 0.067 | 0.276 * (2.493) | 0.111 | −18,637.05 *** (−3.490) | 5339.754 |
| C | 2.730 *** (5.450) | 0.501 | 1.588 * (2.520) | 0.630 | 17,753.54 (0.439) | 40,426.40 |
| Adj R-squared | 0.894 | | 0.124 | | 0.241 | |
| F-statistic | 126.061 *** | | 4.091 * | | 5.739 *** | |

Significant codes: 0 '***' 0.001 '**' 0.01 '*' 0.05, Source: author's computation (2021).

The *t*-test analysis shows that each explanatory variable has a varying effect on the dependent variable. EMS is positively and significantly affected by AVA and RENT. A percentage increase in AVA is associated with a 0.235% rise in EMS. Hereafter, a percentage increase in RENT increases EMS by 0.132% and is statistically significant at a 5% level of significance, ceteris paribus. However, EMS is negatively and significantly affected by EXC. EMS falls by 0.282% when EXC increases by 1%. Two other variables, EXP and DEBT, had no significant impact on EMS. This indicates that Hypotheses 2a, 3a, and 4a are accepted in this study, while Hypotheses 1a and 5a are not.

The second analysis shows that EXC and DEBT are two positive and significant factors on TEMP at a 5% significance level. That means if EXC increases by 1%, TEMP increases by 0.091 °C. Meanwhile, TEMP will rise by 0.276 °C due to a 1% increase in DEBT. The other three variables, EXP, AVA, and RENT, do not influence TEMP. Overall, Hypotheses 3b and 5b were accepted, while Hypotheses 1b, 2b, and 4b were rejected. The last analysis shows that only DEBT has a significant influence on FIRE. A percentage increase in DEBT decreases FIRE by 18,637.05 ha, and is statistically significant at a 1% level of significance, ceteris paribus. EXP, AVA, EXC, and RENT have no significant effect on FIRE. Thus, Hypothesis 5c is accepted, while Hypotheses 1c, 2c, 3c, and 4c are rejected.

## 5. Discussion

### 5.1. Determinant Factors of EMS in Central Asia

Central Asia is one of many regions worldwide that face severe environmental issues. This is closely linked to human activities, mainly mining and agriculture [16,57]. Hence, Kazakhstan and Turkmenistan are the least sustainable compared to other countries in Europe and Central Asia [101], whereas ecological degradation imposes high financial

costs throughout the world, human health is harmed, resource productivity is lost, and ecosystem services are degraded due to environmental carelessness [102].

There are three acceptable Hypotheses in this subsection (2a, 3a, and 4a), but Hypotheses 1a and 5a are not acceptable.

According to the findings of this study, a rise in AVA causes an increase in EMS in Central Asia. This is in accordance with Hypothesis 2a. Similar cases appear to occur in several developed European Union countries. AVA's change is more than 1.41%, resulting in increased pollution in this sector [103]. According to Khan et al. [79], activities that promote higher levels of economic growth, such as AVA, result in higher energy consumption, which contributes to climate change. Saidi and Hammami [104] also stated that carbon emissions result from economic activities. Erokhin et al. [105] said that agriculture in Central Asian countries is competitively poor because it still uses little advanced technology, including environmentally friendly technology. In addition, these countries are also less focused on improving energy efficiency [50].

Actually, each country in Central Asia has policies to limit emissions as the industry grows. For example, Kazakhstan and Tajikistan emphasize upgrading the energy industry, increasing efficiency, and diversifying the industry by providing incentives for renewable energy sources [16]. However, it appears that this does not apply to the agricultural sector, implying that a rise in AVA will still increase EMS. In addition, it is impossible to stop economic activity since it will have several detrimental consequences for human life. Raza et al. [106] show that increasing AVA can reduce $CO_2$ emissions in Pakistan. Hence, the solution is developing renewable energy and innovative environmental technologies [106,107].

A rise in RENT causes an increase in EMS. This is in accordance with Hypothesis 3a. These findings are consistent with Agboola et al. [85], who found a substantial positive connection between Saudi Arabia's total natural resource rent and $CO_2$ emissions in the short and long run. Similar cases also occur in Sub-Saharan African countries [86]. RENT encourages Central Asia to over-exploit the ecosystems' natural resources to meet their needs [51]. Li et al. [108] reinforced it by stating that this situation increases reliance on the environment, puts a lot of strain on natural resources, and makes it challenging to maintain a sustainable ecosystem. This shows that if conservation and management choices are overlooked, so will an over-reliance on RENT harm EMS. However, OECD countries have proven that increasing RENT can still reduce EMS if improving institutional quality focuses on conservation [109].

EXC has a negative relationship with EMS in Central Asia, or inversely with the previous two variables. This is contrary to Hypothesis 4a. Similar findings are seen in Vietnam, where EXC and EMS have a negative association [110]. According to the findings of our study, an increase in the value of EXC (depreciation) causes an increase in the import price of agricultural production factors. Actually, Central Asian countries have long imported agricultural production factors [27]. The usage of agricultural production factors decreases as EXC rises, which has implications for the decline in EMS. Even these agricultural production factors are being replaced by more environmentally friendly resources at cheaper prices, as is often the case with fossil fuel consumption [111].

### 5.2. Determinant Factors of TEMP in Central Asia

Temperature rises in Central Asia must be addressed immediately since they have been linked to the considerable growth of the carbon source area between 2001 and 2008. A rise in temperature causes soil respiration to speed up, reducing carbon sinks in the ecosystem [20]. This is also supported by Han et al. [112]. In the previous three decades, climate data shows that Central Asia has undergone a yearly increase in average temperature and a drop in rainfall. This is in sharp contrast to the fact that Central Asian countries have decided to join the PAC, restricting the surface air temperature to 1.5 °C, relative to 2 °C, which would significantly reduce the frequency of severe heat occurrences in a country [113].

There are two acceptable hypotheses in this subsection (3b and 5b), but Hypotheses 1b, 2b, and 4b are not acceptable.

Based on our findings, a rise in EXC leads to an increase in TEMP. This is in accordance with Hypothesis 3b. This might be in contrast to our prior finding that higher EXC led to lower EMS. However, we point out that the EMS in this study is only applicable to the agricultural sector. This suggests that an increase in EXC might increase EMS in other sectors, leading TEMP to rise in Central Asian countries. EXC fluctuations will spur business growth. Furthermore, this will impact increasing temperature as one of the business's outputs [114]. De Araujo Barbosa et al. [84] also stated that the pressure on the ecosystem has increased along with the currency exchange rate volatility.

A rise in another variable, DEBT, also causes an increase in TEMP. This is in accordance with Hypothesis 5b. The findings of this study coincide with those of Essers et al. [115], who claim that DEBT has a lousy track record in coping with climate change. DEBT is commonly utilized in developing countries to stimulate economic growth or create an infrastructure that can harm the environment [116]. This also happens in Central Asian countries where DEBT seems to help grow economically, but where environmental reforms are overlooked. According to the EKC hypothesis, this is a common occurrence. Even in developed countries such as the G20, debt to the energy sector (about USD 250 billion) is mainly employed for fossil fuels rather than cleaner energy alternatives [117]. Another problem is that, in Central Asian countries, the implementation of DEBT has not been accompanied by an effective environmental management system. According to Han et al. [112], funding connected to the environment in Central Asia is more focused on the forest land use procedure than on the temperature reduction program.

### 5.3. Determinant Factors of FIRE in Central Asia

Forest fires in Central Asia are commonly triggered by converting forests to agricultural land and human settlements [118]. This is thought to significantly contribute to GHG emissions and climate change [119]. There is only one acceptable hypothesis in this subsection (5c), but Hypotheses 1c, 2c, 3c, and 4c are not acceptable.

In our study, increasing DEBT could reduce forest fires in Central Asia. This is contrary to Hypothesis. As stated earlier, DEBT in Central Asia focuses on the forest land use procedure to reduce FIRE [112]. In fact, forest fires in Central Asia have reduced from 2001 to 2019 [57]. For example, Kazakhstan was the most-affected by fire in Central Asia. Forest fires were mainly seen in Kazakhstan's northern and eastern regions. Currently, forests previously harmed by fires are undergoing gradual restoration [57]. It seems that the issuance of DEBT in Central Asia has been followed by international environmental sustainability programs, such as the reduction of deforestation and degradation, the conservation of forest carbon stocks, the sustainable management of forests, and the enhancement of forest carbon stocks [119]. This is an excellent condition for Central Asia since forests are a giant carbon sink in a changing climate and absorb a large portion of global terrestrial carbon [112].

In our findings, exports did not influence environmental sustainability in Central Asia, regardless of the other explanatory factors employed in this study. The export market appears to have failed to offer enough and suitable environmental protection. Large-market countries must pay more attention to imports and establish product requirements or rules, including environmental, health, and safety rules [102].

### 5.4. Our Findings, the EKC Hypothesis, and the PAC

Our findings reveal that EG has sound and harmful environmental effects in Central Asian countries. However, a closer examination reveals that EG can have a more significant harmful influence on the ecosystem in Central Asia. This finding supports the EKC hypothesis, which states that economic activity has a detrimental influence on the environment.

As shown in Figure 2, regarding the hypothetical EKC cycle, we assume Central Asian countries are now in stage 2 (scale effect of the industrial economy). At this time, all-natural resources are being fully used for economic growth and income. There has been a

paradigm shift in economic development, moving from a pollution-free agricultural sector to a polluted industrial sector. The ecosystem aspect has not become the primary focus. As a result, $CO_2$ emissions and temperatures have risen across Central Asia. This result also serves as a cautionary note for the PAC's and SDGs' accomplishments in Central Asia.

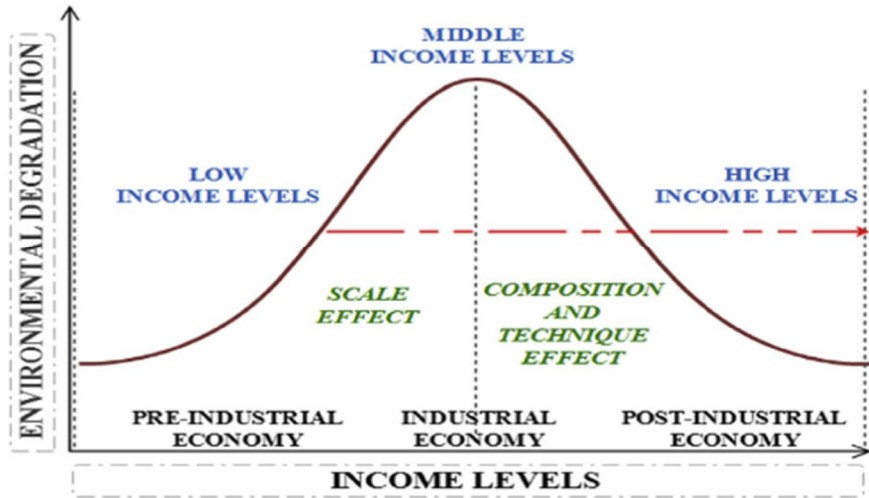

**Figure 2.** Schematic of inverted U-shaped EKC hypothesis (source [120]).

In our opinion, there are several causes for this problem, including:

1.  Central Asian countries have not used advanced technology [105]. Many countries face this problem because they rely on simple technologies to extract resources and produce goods. Worse, the technology is not being utilized efficiently;
2.  There is a lack of community and public sustainability education [58]. As a result, they are unconcerned with environmental degradation and its consequences for human life. In fact, this is detrimental to human health and causes high financial costs worldwide [102];
3.  There is a low level of commitment to eco-friendly policies. Environmental issues receive less attention because the government and society are more concerned with economic development. Even Central Asian countries have environmental policies, but they are not well implemented [16];
4.  There is a lack of worldwide support for environmental protection. This may be observed in each country's low level of commitment globally to implement the COP21 outcomes. Likewise, there is a lack of recognition and actions from everyday citizens in their lives (e.g., recycling);
5.  As the hypothetical EKC cycle shows, many of a country's industries must attain economies of scale before shifting into a sustainable economy. Otherwise, developing countries would not cover the initial costs of a multi-year transition to a more environmentally friendly and sustainable economy.

However, a sense of optimism must still be built for environmental improvement in Central Asia. Several explanatory variables can reduce environmental degradation. This indicates that EG in Central Asia is ready to pass the EKC hypothesis' turning point. Bibi and Jamil [121] stated that EG would improve environmental quality in Central Asia in the future. We need to ensure that the Central Asian government takes the necessary actions (which we discuss in Section 6.2) to ensure that this prediction comes true in the future. This is necessary because several countries have failed to enhance environmental quality and economic growth. For example, the truth of the EKC hypothesis was not proven in South Africa [122]. Even in some OECD and non-OECD countries (such as Africa, Asia, and Latin America), the EKC curve is N-shaped [123]. Another key point to remember is that each country's approach to environmental sustainability can be different [124]. This

should consider each country's unique conditions. Tendencies such as these can be a guide for Central Asian countries to achieve their SDGs and PAC.

## 6. Conclusions, Implications, Limitations, and Future Research

### 6.1. Conclusions

EG accelerates very fast in Central Asia. Hence, many problems exist in Central Asia's ecosystem. Our study shows that EG in Central Asian countries runs according to the EKC hypothesis, and it is now in the second stage (the industrial economy). Overall, Hypotheses 2a, 3a, 3b, 4a, 5b, and 5c are confirmed, while the other hypotheses are rejected. EG has both positive and negative impacts on environmental sustainability in Central Asia. The increase in agriculture forestry and fishing value-added and total natural resource rents leads to increased $CO_2$ emissions from on-farm energy use. On the other hand, an increase in exchange rates can mitigate $CO_2$ emissions from on-farm energy use. Another indicator of environmental sustainability in our study, temperature change, will increase in line with the increase in exchange rates and external debt stocks. Finally, a rise in external debt stocks will reduce Central Asian forest fires.

Our findings show that EG in agriculture in developing countries, particularly in the Central Asian region, has resulted in environmental harm. Previous studies provided in this article also show that EG can stimulate economic growth while also negatively influencing the environment. However, $CO_2$ emissions and temperature rise due to the employment of environmentally unfriendly chemical production factors and technologies. Unfortunately, many of these technologies are still traditional, resulting in water, soil, and air pollution. This contributes to the advancement of knowledge in the area of improved EG management, which are intended to have an excellent economic and environmental effect.

### 6.2. Implications

Based on this study, we recommend several strategies to meet SDGs and the PAC. First, Central Asian countries require a path for achieving the SDGs and the PAC. The roadmap must include various activities, environmental damage prediction, and environmental mitigation and adaptation strategies related to agriculture and EG. This roadmap must be disseminated to the micro-level so that the community (mainly farmers) may engage in environmental sustainability initiatives. Second, partnerships with investors, governments, and researchers from other countries should be improved to increase energy efficiency, renewable energy sources, innovative environmental technologies, and environmental research. It has been proven that increasing energy efficiency boosts economic productivity and growth. Meanwhile, research collaboration will facilitate the discovery or transfer of environmentally friendly technologies from developed countries to Central Asian countries. Third, regional and international support is being applied to Central Asian countries to strengthen environmentally damaging farming practice regulations. The support takes the form of transferring education and funding to Central Asian countries to mitigate the adverse effects of climate change on the global economy.

### 6.3. Limitations and Future Research

As researchers, we believe this study still has some limitations. First, there are only a limited number of explanatory variables. Meanwhile, EG has several variables that were not considered in this study. Hence, we recommend some variables that represent EG include foreign direct investment, trade agreements, trade duties, world agricultural products or oil prices and others. Second, we used a simple data analysis method, namely, static panel data analysis. This method has shortcomings, including serial correlation and heteroscedasticity issues [125]. This may result in spurious regression findings. Even though we have made every effort to reduce the occurrence of such findings, there is still a chance that spurious regression will occur. Thus, we propose applying the generalized method of moment (GMM) and systems GMM to overcome the shortcomings of static panel data analysis. Finally, our study only focuses on agriculture. We recommend further

study on the impact of EG on various sectors in Central Asia. This is expected to review the impact of EG broadly and follow the EKC hypothesis in more detail.

**Author Contributions:** Conceptualization, A.D.N., M.F.-F. and Z.L.; methodology, A.D.N.; software, A.D.N.; validation, A.B., A.D.N., M.F.-F. and Z.L.; formal analysis, A.D.N.; investigation, M.F.-F. and Z.L.; writing—original draft preparation, A.B. and A.D.N.; writing—review and editing, A.B. and A.D.N.; supervision, M.F.-F. and Z.L. All authors have read and agreed to the published version of the manuscript.

**Funding:** This research received no external funding.

**Institutional Review Board Statement:** Not applicable.

**Informed Consent Statement:** Not applicable.

**Data Availability Statement:** The data presented in this study are available on request from the corresponding author.

**Acknowledgments:** We would like to thank the reviewers who gave us suggestions on how to develop this article.

**Conflicts of Interest:** The authors declare no conflict of interest.

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
