# Peer review of "Global Challenges and Responses: Agriculture, Economic Globalization, and Environmental Sustainability in Central Asia"

_sustainability, doi:10.3390/su14042455_

Round 1
Reviewer 1 Report
63 aimed rather than aims? Tense in the whole para could be revisited
107 omit bothe the and commodity
122 omit in terms of the environment
123 change if to by
124 omit still
133 after will insert possibly or likely
132 aims not aimed
144 omit Meanwhile,
153 unclear argument; para needs reworking for clarity
159 replace is by was at the turn of the century
164 States not Stats
167 insert maintain before authoritarian
169 replace this by such abundance
176 insert as to before whether and replace American by foreign
223 model does not make sense
229 mentioning a classic article without citing it does not seem like good practice:
Economic Growth and the Environment
Gene M. Grossman and Alan B. Krueger
Quarterly Journal of Economics
Vol. 110, No. 2 (May, 1995), pp. 353-377
https://doi.org/10.2307/2118443
242 declines not declined
245 change material to topic
286 use of quintile here does not make any sense to this reader
286-97 throughout causality is unclear or ambiguoue
311-21 similar causality ambiguity
336 are not is
426 differenced?
461 omit Unfortunately
553 insert the before most
570 change has to can have
612 replace Principles like this by Tendencies such as these
642ff at last some belated humility; taken literally the journal should best await a more definitive submission!
Author Response
Dear Professor
Thank you so much for your efforts, suggestions, and comments. We feel that as a result of this, our manuscript will be better, systematic, and achieve high quality. We believe all of this has been done to ensure that our manuscript meets the expectations of the Sustainability journal.
Here is our response to your comments and suggestions
- Reviewer: Tense in the whole para could be revisited
Author: Based on the Professor suggestion, we have corrected the grammar and some words (attached)
2. Reviewer: mentioning a classic article without citing it does not seem like good practice: Economic Growth and the Environment (Gene M. Grossman and Alan B. Krueger)
Author: We have included the publication proposed by the Professor
3. Reviewer: throughout causality is unclear or ambiguous
Author: We have corrected some parts of our manuscript so that our hypothesis is no more ambiguous (attached)
We hope that all of this has met your expectations.
Thank you
Best regards

Reviewer 2 Report
The paper is a well-done study, logically structured and having proper application of the methods as well as their interpretation and discussion of the limitations of the work.
I would like to advise the authors to introduce more articulated research problem and research question at the beginning of the paper, as well as linking it to the literature review, theoretical framework and put forward hypotheses.
Author Response
Dear Professor
Thank you so much for your efforts, suggestions, and comments. We feel that as a result of this, our manuscript will be better, systematic, and achieve high quality. We believe all of this has been done to ensure that our manuscript meets the expectations of the Sustainability journal.
Here is our response to your comments and suggestions
- Reviewer: The paper is a well-done study, logically structured and having proper application of the methods as well as their interpretation and discussion of the limitations of the work.
Author: Thank you Professor for your compliments on our manuscript
2. Reviewer: I would like to advise the authors to introduce more articulated research problem and research question at the beginning of the paper, as well as linking it to the literature review, theoretical framework and put forward hypotheses.
Author: Based on the Professor's suggestion, we are trying to add research problems (lines 137-140) and research questions (lines 141-143). We also improve the literature review and corrected the theoretical framework (line 283-306) and hypotheses to make our manuscript more systematic
We hope that all of this has met your expectations.
Thank you
Best regards

Reviewer 3 Report
- Although the scope of the study was well-defined in the Abstract (lines 13 to 15), the purpose of conducting the analysis remains undisclosed. Main conclusions should be also displayed in this section.
- A broad range of arguments were provided in the Introduction, but their connection is scant. The thread between agriculture, economic globalization (EG) and environmental sustainability (ES) should be therefore enhanced, for instance by linking all notions through an in-depth literature review. However, literature review is mostly descriptive based on the characterization of EG and ES in the selected Asian countries.
- Selection criteria of explanatory and dependent variables are unknown. Same for the given panel models and statistical tests.
- Figure 1 illustrates no theoretical framework, but some variables considered in the study. Are the 15 formulated hypotheses covered by the EKC hypothesis proposed by Grossman & Krueger? If so, what is the scientific contribution of the study?
- Discussion should be focused on the 15 hypotheses posed.
- Due to the lack of support based on findings, author´s opinions gathered in subsection 5.4 can be deemed as arbitrary.
- Neither theoretical nor practical implications were provided, but subsection 6.2. encompasses some recommendations. Consequently, the contribution of the study in the field of knowledge is unknown.
Several statistical analyses were mostly conducted taking as reference a set of given variables to examine the evolution of 6 Asian countries in the 1994-2019 period. Beyond statistics provided, the purpose remains vague that is why implications of the study are controversial.
Author Response
Dear Professor
Thank you so much for your efforts, suggestions, and comments. We feel that as a result of this, our manuscript will be better, systematic, and achieve high quality. We believe all of this has been done to ensure that our manuscript meets the expectations of the Sustainability journal.
Here is our response to your comments and suggestions
- Reviewer: Although the scope of the study was well-defined in the Abstract (lines 13 to 15), the purpose of conducting the analysis remains undisclosed. Main conclusions should be also displayed in this section.
Author: Based on the Professor suggestion, the abstract has been improved making the aims clear (line 13-15) and displaying the main conclusion (line 22-27)
2. Reviewer: A broad range of arguments were provided in the Introduction, but their connection is scant. The thread between agriculture, economic globalization (EG) and environmental sustainability (ES) should be therefore enhanced, for instance by linking all notions through an in-depth literature review. However, literature review is mostly descriptive based on the characterization of EG and ES in the selected Asian countries.
Author: We have added the sentence to explain the relationship between economic globalization and environmental sustainability clearly (line 137-140 and line 283-306). We also corrected the literature review (lines 159-258).
3. Reviewer: Selection criteria of explanatory and dependent variables are unknown. Same for the given panel models and statistical tests.
Author: Based on the Professor's comment, we add justification for selecting explanatory and dependent variables in the theoretical framework section (line 283-306) and hypotheses (line 314-388) for improving our manuscript. We've also corrected Figure 1's theoretical framework and add the reasons for choosing panel analysis in the method section (line 405-406 and 441-444).
4. Reviewer: Figure 1 illustrates no theoretical framework, but some variables considered in the study. Are the 15 formulated hypotheses covered by the EKC hypothesis proposed by Grossman & Krueger? If so, what is the scientific contribution of the study?
Author: Based on the Professor's comment, we've corrected Figure 1's theoretical framework and provide an explanation of the relationship (line 283-306). We have added the scientific contribution of this study to the conclusion section (lines 729-739).
5. Reviewer: Discussion should be focused on the 15 hypotheses posed.
Author: Based on the Professor's suggestion, we improve the discussion section based on the hypothesis so this manuscript is more organized.
6. Reviewer: Due to the lack of support based on findings, author´s opinions gathered in subsection 5.4 can be deemed as arbitrary.
Author: Based on the Professor's suggestion, we improve the theoretical framework, hypothesis, and discussion to support subsection 5.4.
7. Reviewer: Neither theoretical nor practical implications were provided, but subsection 6.2. encompasses some recommendations. Consequently, the contribution of the study in the field of knowledge is unknown.
Author: We display the scientific contribution of this study in the conclusion section (lines 729-739).
8. Reviewer: Several statistical analyses were mostly conducted taking as reference a set of given variables to examine the evolution of 6 Asian countries in the 1994-2019 period. Beyond statistics provided, the purpose remains vague that is why implications of the study are controversial.
Author: Based on the Professor"s comment, we have improved the systematics of the manuscript (research questions, aims, theoretical framework, hypotheses, and discussions) to make our manuscript more focused.
We hope that all of this has met your expectations.
Thank you
Best regards
Reviewer 4 Report
While the paper attempts to look at an interesting topic, there needs to be major improvements.
- Provide clear justification the use of the sample of countries in the analysis.
- Provide theoretical links before establishing the hypothesis. For example, what is the theoretical link between external debt stock and CO2 emissions.
- Provide summary of data, that is, descriptive statistics and correlation coefficients.
- Review model specification. Are all variables in the linear models (2), (4), and (6) in log-form (dependent and independent variables)?
- On page 8, define theta (see lines 376-379).
- On page 9, define rho (lines 388-389)
- On page 9, define betas (lines 404-405)
- The paper notes that, "five non-stationary variables were first differencing into stationary variables". How were these differenced variables estimated in the linear regression model? That is, were all first differenced quantities positive to justify log-form?
- Also, how can the results be interpreted as percentage change when the first-difference variable are used, for those which were non-stationary in their levels?
- Following on point 9, and looking at the model specification, the only the independent variables are expressed in log-form (and not the dependent variables). How can then the findings be interpreted as percentage change. We can only do so if both dependent and independent are expressed in log forms in their levels.
- Another important point is that the paper does not directly examine EKC because there was not threshold effect that were analyzed. Discussion seems a bit ambitious.
Author Response
Dear Professor
Thank you so much for your efforts, suggestions, and comments. We feel that as a result of this, our manuscript will be better, systematic, and achieve high quality. We believe all of this has been done to ensure that our manuscript meets the expectations of the Sustainability journal.
Here is our response to your comments and suggestions
- Reviewer: Provide clear justification the use of the sample of countries in the analysis.
Author: Based on the Professor's comment, we add the justification for choosing six Central Asian countries as samples of our manuscript in the introduction (line 83-108) and methods section (line 393-395)
2. Reviewer: Provide theoretical links before establishing the hypothesis. For example, what is the theoretical link between external debt stock and CO2 emissions.
Author: Based on the Professor's suggestion, we have improved the theory in figure 1 and its explanation (line 283-306).
3. Reviewer: Provide summary of data, that is, descriptive statistics and correlation coefficients.
Author: Based on the Professor's suggestion, we have added descriptive statistics in Table 1 and correlation coefficients in Table 2.
4. Reviewer: Review model specification. Are all variables in the linear models (2), (4), and (6) in log-form (dependent and independent variables)?
Author: Dear Professor, we use semi-log forms in equations (2), (4), and (6). The dependent variable in model 2 uses the log form. While the dependent variable models 4 & 6 do not use the log form. While the independent variables exp, ava, exc, and debt are in log form and the rent is not in log form either in models 2, 4, and 6.
5. Reviewer: On page 8, define theta (see lines 376-379).
Author: We have defined theta on line 449
6. Reviewer: On page 9, define rho (lines 388-389)
Author: We have defined rho on line 463
7. Reviewer: On page 9, define betas (lines 404-405)
Author: We have defined beta on line 475
8. Reviewer: The paper notes that, "five non-stationary variables were first differencing into stationary variables". How were these differenced variables estimated in the linear regression model? That is, were all first differenced quantities positive to justify log-form?
Author: Based on the Professor's comment, we have added publications (Panel Data and Models of Change: A Comparison of First Difference and Conventional Two-Wave Models) to strengthen the argument that differenced variables can be estimated in the linear regression model. We have added this argument in the method section (lines 441-444). We have also added reasons and publication (from Gujarati) regarding the importance of the log form to produce the best estimation results (line 417).
9. Reviewer: Also, how can the results be interpreted as percentage change when the first-difference variable are used, for those which were non-stationary in their levels?
Author: Dear Professor, we use log form for some variables (semi-log) so the interpretation of this variable in percentage form. All variables that we use in the model are stationary at level or first difference. As far as we understand, it does not matter the variable in the log form is differentiated.
10. Reviewer: Following on point 9, and looking at the model specification, the only the independent variables are expressed in log-form (and not the dependent variables). How can then the findings be interpreted as percentage change. We can only do so if both dependent and independent are expressed in log forms in their levels.
Author: Dear Professor, in model 2, the dependent variable in log form, hence the interpretation must be expressed in percentages. We do not utilize the dependent variable in log form in models 4 and 6, therefore the interpretation is based on the original size.
11. Reviewer: Another important point is that the paper does not directly examine EKC because there was not threshold effect that were analyzed. Discussion seems a bit ambitious.
Author: Based on Professor suggestion, we have improved the systematics of the manuscript (research questions, aims, theoretical framework, hypotheses and discussions) to make our manuscript more focused.
We hope that all of this has met your expectations.
Thank you
Best regards
Round 2
Reviewer 3 Report
Please arrange numbering of references from the 37th one to the 125th
Author Response
Dear Professor
I hope you are doing well and in good health
We have corrected the reference according to your suggestion. Thank you for your attention to our manuscript. Now we are more confident that our manuscript has improved for the better and can contribute to the development of knowledge.
Thank You
Reviewer 4 Report
Overall, the authors have made some genuine attempts to address my concerns. I congratulate the authors for their efforts.
Author Response
Dear Professor
I hope you are doing well and in good health
Thank you for your attention to our manuscript. Now we are more confident that our manuscript has improved for the better and can contribute to the development of knowledge.
Thank You
Best wishes